Integrative bioinformatics analysis identifies C1orf198 as a novel prognostic biomarker in colorectal cancer

Yang Changjiang 1 2
Geng Xuhua 1
Zhao Zihan zzh960407@pku.edu.cn 1911210739@pku.edu.cn 1
1 Aerospace Center Hospital, Peking University Aerospace School of Clinical Medicine , Beijing , China
2 Department of Gastroenterological Surgery, Peking University People’s Hospital , Beijing , China
Ozdag Hilal
Electronic publication date: 2025 Oct 21
Publication date: 2025
Volume: 13
Electronic Location ID: e20227
Received 2025 May 16; Accepted 2025 Sep 22
Copyright: ©2025 Yang et al.
Copyright year: 2025
Copyright holder: Yang et al.
License: This is an open access article distributed under the terms of the Creative Commons Attribution License, which permits unrestricted use, distribution, reproduction and adaptation in any medium and for any purpose provided that it is properly attributed. For attribution, the original author(s), title, publication source (PeerJ) and either DOI or URL of the article must be cited.
License URL: https://creativecommons.org/licenses/by/4.0/

Keywords: C1orf198, Colorectal cancer, Immune infiltration, Prognosis, Metastasis

Funding: The authors received no funding for this work.

==============================
Colorectal cancer (CRC) is a leading cause of cancer-related mortality, necessitating the identification of novel prognostic biomarkers to improve patient management. In this study, we integrated bioinformatics analyses and experimental validation to explore the role of C1orf198 in CRC. Data from The Cancer Genome Atlas (TCGA) and Gene Expression Omnibus (GEO) revealed significantly upregulated C1orf198 mRNA in CRC tissues compared to normal counterparts, confirmed by immunohistochemistry (IHC) in clinical samples. High C1orf198 expression correlated with advanced tumor stages (T, N, M) and poor survival outcomes, including shorter overall survival (OS), disease-specific survival (DSS), and progression-free interval (PFI). Functional enrichment analyses highlighted involvement in extracellular matrix organization, cell adhesion, and oncogenic signaling pathways such as PI3K-AKT and focal adhesion. Immune infiltration analysis showed positive correlations with stromal/immune scores and M2 macrophage infiltration, linking C1orf198 to tumor microenvironment (TME) remodeling. Notably, C1orf198 was strongly associated with cytokines CXCL12 and receptors CXCR5, which mirrored its immune cell correlations. Collectively, our findings identify C1orf198 as a potential prognostic biomarker in CRC, implicating its role in TME modulation and oncogenic progression.

Introduction

Colorectal cancer (CRC), ranked among the most common malignant tumors globally, poses a significant threat to human health (Dekker et al., 2019). Recent evidence has highlighted the emergence of new favorable subsets within cancers of unknown primary (CUP), including CRC-related CUP. This distinct clinical entity is managed as CRC and has contributed to the observed increase in current CRC incidence (Rassy et al., 2020). Although considerable advancements have been achieved in diagnostic and therapeutic approaches, treatment outcomes for a substantial portion of CRC patients (particularly elderly individuals) still remain suboptimal (Baidoun et al., 2021). Notably, in the context of locally advanced disease, older patients with T4 CRC are more susceptible to severe postoperative complications; however, age itself does not independently dictate survival outcomes. Instead, their prognosis is often confounded by differences in stage at presentation, tumor site, preexisting comorbidities, and the type of treatment received, highlighting the need for personalized management strategies in this population (Osseis et al., 2022). The discovery of novel biomarkers and clarification of the underlying molecular mechanisms driving CRC progression are pivotal for enhancing clinical management and improving patient prognosis (Li et al., 2021).

The study of the association between uncharacterized proteins and tumors has gradually become a hotspot in the field of tumor biology. These proteins account for over 10% of the genome, and their functions remain unclear (Duek et al., 2021). However, increasing evidence suggests that they play critical roles in tumor initiation, progression, and therapeutic resistance (Ma et al., 2019). C1orf198, an understudied gene located on chromosome, has been associated with cellular mechanisms including chromatin organization and immune regulation. However, its involvement in cancer development remains poorly understood. Previous studies have suggested that C1orf198 plays a role in the initiation and progression of gastric (Wang et al., 2017) and breast cancers (Lee et al., 2018), but its relationship with CRC remains unclear. On the other hand, emerging evidence suggests that genes involved in tumor microenvironment (TME) remodeling, including extracellular matrix (ECM) components and immune cell crosstalk, are critical for CRC progression. Dysregulation of cell adhesion molecules, ECM-receptor interactions, and immune microenvironment has been linked to metastasis and therapeutic resistance in CRC. Notably, emerging evidence highlights microRNAs (miRNAs) as key regulators of drug resistance in this context: FOLFOX resistance in advanced CRC is significantly associated with upregulation of miR-19a, while resistance to anti-VEGF agents correlates with miR-126 upregulation. Similarly, resistance to anti-EGFR inhibitors is linked to overexpression of miR-31, miR-100, miR-125b, and downregulation of miR-7, underscoring the multifaceted role of non-coding RNAs in modulating treatment response (Boussios et al., 2019).

Here, we hypothesized that C1orf198 contributes to CRC pathogenesis by influencing TME components and oncogenic signaling. Using multi-omics datasets and experimental validation, we aimed to investigate C1orf198 expression patterns, clinical correlations, functional pathways, and immune infiltration associations in CRC. Identifying C1orf198 as a prognostic biomarker could provide insights into novel therapeutic targets for CRC.

Materials and Methods

Acquisition and processing of data

RNA-seq data along with matched clinical profiles from CRC tumors and paired adjacent normal tissues were obtained from The Cancer Genome Atlas (TCGA, https://portal.gdc.cancer.gov/), a prominent public cancer genomics resource. Additionally, the datasets GSE113513, GSE89076, GSE22598, and GSE110224 utilized in this study were retrieved from the Gene Expression Omnibus (GEO), a widely accessed functional genomics repository that supports MIAME-compliant data submissions.

Patient samples and clinical specimens

Supplied by Shanghai Outdo Biotech Company (Shanghai, China), the colorectal cancer (CRC) tissue microarray (HColA160CS01) comprised 80 paired tumor and adjacent normal tissue samples. Ethical approval for the study protocols was granted by the company’s Ethics Committee under the approval ID: YB M-05-02. All participants provided written informed consent prior to their involvement in the study.

Immunohistochemistry staining protocol

Processing of CRC and normal tissue sections followed this protocol: after embedding in paraffin, specimens underwent deparaffinization with dimethylbenzene and sequential rehydration through ethanol gradients. Antigen retrieval was performed using a sodium citrate buffer via microwave treatment at 95 °C. Endogenous peroxidase activity was blocked by incubating specimens in 3% hydrogen peroxide for 10 min. To reduce non-specific binding, tissues were then incubated for one hour in a blocking solution containing 10% fetal bovine serum. The primary antibody against C1orf198 (1:100 dilution, CST #33418) was applied and incubated overnight at 4 °C. After washing, sections were treated with an HRP-conjugated anti-rabbit secondary antibody in the dark. Immunostaining was visualized using DAB, followed by counterstaining with hematoxylin. Slides were dehydrated, mounted, and prepared for microscopic evaluation.

C1orf198 expression was assessed using two criteria: the percentage of positively stained cells and staining intensity. Positively stained cells were categorized into four groups based on their proportion: category 0 (0–10%), category 1 (10–40%), category 2 (40–70%), and category 3 (>70%). Staining intensity received numerical scores: 1 for weak, 2 for moderate, and 3 for strong visibility. The final immunohistochemistry (IHC) score for each sample was calculated by summing the cell positivity percentage score and the intensity score, resulting in a combined score ranging from 0 to 6. Samples were classified as low expression (0–3 points) or high expression (4–6 points). All procedures were conducted in strict accordance with standardized guidelines to ensure methodological consistency and regulatory compliance.

Survival analysis

Survival outcomes, including overall survival (OS), disease-specific survival (DSS), progression-free interval (PFI), and disease-free survival (DFS), were evaluated using the Kaplan–Meier method with log-rank testing. Patients were stratified into low- and high-expression groups based on the median C1orf198 expression level as the cutoff threshold. Cox proportional hazards regression models were applied to assess associations between clinical/pathological characteristics and prognostic endpoints, incorporating these survival metrics into the analysis.

Functional enrichment analysis

Differential gene expression analysis between C1orf198 low and C1orf198 high tissues was performed with significance thresholds set as Fold change >1.5 and FDR < 0.05 to identify DEGs (differentially expressed genes). The ClusterProfiler package (R v3.6.3) was utilized to conduct Gene Ontology (GO) and Kyoto Encyclopedia of Genes and Genomes (KEGG) pathway enrichment analyses, alongside gene set enrichment analysis (GSEA). GO annotations were categorized into molecular functions (MF), cellular components (CC), and biological processes (BP). For GSEA, pathway enrichment was evaluated using normalized enrichment scores (NES) and adjusted p-values, with reference gene sets derived from c2.cp.kegg.v2022.1.Hs.symbols.gmt (KEGG) and c5.go.all.v2022.1.Hs.symbols.gmt (GO) databases. Pathways were considered significantly enriched when meeting criteria of adjusted p < 0.05 and FDR < 0.25.

Immune infiltration analysis

The ESTIMATE algorithm was utilized to compute immune and stromal scores for CRC samples. We leveraged the GSVA package in R to perform single-sample gene set enrichment analysis (ssGSEA), aiming to explore literature-supported associations between SPOCD1 and the hallmark gene signatures of 24 distinct immune cell types (Bindea et al., 2013). The CIBERSORT was also applied to examine the link between C1orf198 and infiltering immune cells. Spearman’s correlation analysis was conducted to examine the association between C1orf198 expression levels and the degrees of immune cell infiltration. Differences in immune cell composition across the low- and high-expression groups of C1orf198 were evaluated using the Wilcoxon rank-sum test.

Statistical methods

Data from TCGA were analyzed using R (v4.2.1; R Core Team, 2022). To compare C1orf198 expression levels between tumor and normal tissues, Wilcoxon rank-sum tests (for independent samples) and signed-rank tests (for paired samples) were applied. Associations between C1orf198 expression and clinicopathological characteristics were evaluated using Welch’s one-way ANOVA, followed by Bonferroni post-hoc tests for multi-group comparisons (or t-tests for two-group analyses). Pearson’s chi-square test was used to assess correlations between C1orf198 expression and clinical factors, with Fisher’s exact test employed for small sample sizes to ensure analytical validity.

The prognostic value of C1orf198 was evaluated using Kaplan–Meier curves. All statistical tests were two-tailed, with significance defined as P ≤ 0.05 to maintain consistency in inferential analysis.

Results

CRC is characterized by upregulated C1orf198 mRNA and protein level

The mRNA and protein expression levels of C1orf198 were analyzed in pan-cancer and CRC tissues. Using TCGA data, comparative analysis of C1orf198 mRNA expression in pan-cancer and normal tissues revealed expression differences across multiple cancer types (Figs. 1A, 1B). TCGA-based box-plot and paired-sample analysis showed that C1orf198 mRNA expression was significantly up-regulated in CRC tissues compared with normal tissues (Figs. 1C, 1D). To comprehensively evaluate the diagnostic utility of C1orf198 in CRC, a ROC analysis was meticulously conducted (Fig. 1E). This outcome implies that C1orf198 has potential as a diagnostic biomarker for CRC, capable of partially differentiating tumor-bearing states from normal conditions. Analyses of GEO datasets (GSE113513, GSE89076, GSE22598, GSE110224) confirmed the up-regulation of C1orf198 mRNA in CRC versus normal samples (Figs. 1F–1I). Subsequently, we performed IHC staining on clinical samples of CRC. IHC score analysis indicated that C1orf198 protein expression was significantly higher in CRC than in normal tissues (p < 0.001) (Fig. 1J). Representative immunohistochemistry images visually corroborated the expression discrepancy between normal and CRC tissues (Fig. 1K). Collectively, these results demonstrate that C1orf198 is prominently overexpressed at both mRNA and protein levels in CRC tissues, suggesting its potential role in CRC tumorigenesis.

Figure 1 The levels of C1orf198 protein and mRNA expression in pan-cancer and CRC relative to normal samples.

(A, B) Comparative analysis of C1orf198 mRNA expression in pan-cancer and normal tissues using TCGA. (C, D) TCGA-based box-plot and paired-sample analysis showing that C1orf198 mRNA expression was up-regulated in CRC tissues versus normal tissues. (E) ROC analysis evaluating the diagnostic value of C1orf198 in CRC. (F–I) Analyses of GEO datasets (GSE113513, GSE89076, GSE22598, GSE110224) revealed up-regulation of C1orf198 mRNA expression in CRC compared with normal samples. (J) IHC score analysis showing that C1orf198 protein expression was up-regulated in CRC compared with normal samples. (K) Representative immunohistochemistry images of C1orf198 in normal and CRC tissues (ns denotes no significance, * p < 0.05, ** p < 0.01,*** p < 0.001).

Correlation between C1orf198 expression and clinicopathologic parameters and prognosis in CRC

Based on the TCGA database, the correlations between C1orf198 expression and clinicopathologic parameters as well as prognosis in CRC were analyzed. Initially, the association between C1orf198 expression and clinicopathologic parameters was explored. Results demonstrated that C1orf198 expression showed no significant association with age or gender, but was significantly correlated with T stage, N stage, M stage, pathologic stage, OS event, DSS event, and PFI event (Figs. 2A–2I). Further Kaplan–Meier analysis based on TCGA data revealed that up-regulated C1orf198 expression was associated with shortened OS, DSS, and PFI times (Log-rank P = 0.004, 0.002, <0.001) (Figs. 2J–2L). Additionally, Kaplan–Meier curves from GEO datasets (GSE17536 and GSE14333) indicated that up-regulated C1orf198 expression was related to reduced DSS/DFS times (Log-rank P = 0.001, <0.001, <0.001) (Figs. 2M–2O). These findings suggest that high C1orf198 expression may be associated with poor prognosis in CRC.

Figure 2 TCGA database-based correlations between C1orf198 expression and clinicopathologic parameters and prognosis in CRC.

(A–I) Analysis of the association between C1orf198 expression and clinicopathologic parameters in CRC using TCGA data, demonstrating associations with age (A), gender (B), T stage (C), N stage (D), M stage (E), pathologic stage (F), OS event (G), DSS event (H), and PFI event (I) (ns represents no significance, *p < 0.05, ** p < 0.01, *** p < 0.001). (J–L) Kaplan–Meier curves based on TCGA data showing that up-regulated C1orf198 expression was associated with reduced OS, DSS, and PFI times. (M–O) Kaplan–Meier curves from GEO datasets (GSE17536 and GSE14333) indicating that up-regulated C1orf198 expression was related to reduced DSS/DFS times.

Biological correlation and pathway analysis of C1orf198 in CRC

To further investigate the biological functions of C1orf198 in CRC, we analyzed its co-expressed genes and enriched functional pathways. We first selected the top 20 genes most significantly correlated with C1orf198 expression (Fig. 3A), visualized in a heatmap, to explore the molecular mechanisms of C1orf198-mediated biological processes in CRC. Figure 3C presents a volcano plot highlighting DEGs between low- and high-C1orf198-expression subgroups, where red and blue dots denote genes upregulated or downregulated in the high-expression group, respectively. This analysis identified 1,151 upregulated and 690 downregulated genes, forming the basis for subsequent functional enrichment. Co-expressed gene analysis (Fig. 3B) further linked C1orf198 to processes like “myeloid cell differentiation”, “regulation of epithelial cell proliferation” and “regulation of small GTPase mediated signal transduction,” while GO and KEGG pathway analyses of these upregulated genes revealed significant enrichment in biological processes such as “extracellular matrix organization”, “extracellular structure organization” and “cell junction assembly”, “cell–cell adhesion via plasma-membrane adhesion molecules”, cellular components such as “cell–cell junction”, “collagen trimer” and “collagen-containing extracellular matrix”, as well as molecular functions like “integrin binding” and “extracellular matrix structural constituent”. KEGG pathways including “ECM-receptor interaction,” “PI3K-AKT signaling,” “ECM-receptor interaction”, and “Focal adhesion” were prominently enriched, suggesting their involvement in tumor microenvironment remodeling and oncogenic signaling (Fig. 3D). GSEA (Figs. 3E–3G) further illuminated distinct functional landscapes: Hallmark gene sets (Fig. 3E) showed enrichment in “Angiogenesis” and “Epithelial-Mesenchymal Transition (EMT),” aligning with pro-tumor microenvironment remodeling. GO-related gene sets (Fig. 3F) emphasized “Wnt signaling pathway” and “integrin/collagen-mediated signaling,” while KEGG GSEA (Fig. 3G) underscored activation of “Calcium signaling” and “Focal adhesion” pathways. These integrative analyses clarify that C1orf198 influences CRC progression through coordinated regulation of ECM-adhesion networks, oncogenic signaling (e.g., PI3K-AKT, Wnt signaling pathway), and tumor vasculature formation, with potential roles in EMT and microenvironmental crosstalk.

Figure 3 Comprehensive analysis of C1orf198-correlated gene expression, functional enrichment, and pathway associations in CRC.

(A) Heatmap depicting the top 20 genes in CRC with significant correlation (positive) to C1orf198, where red and blue represent high and low expression-related correlation levels, respectively. (B) GO analysis of co-expressed genes with C1orf198, illustrating biological processes and molecular functions associated with these co-expressed genes. (C) Volcano plot showing DEGs between low and high C1orf198 expression subgroups in CRC. Red denotes up-regulated genes in high C1orf198 expression, while blue represents down-regulated genes. (D) GO and KEGG Pathway enrichment analyses for up-regulated and down-regulated DEGs, highlighting significantly enriched biological pathways and functions. (E–G) GSEA plots. (E) Hallmark gene sets, (F) GO-related gene sets, and (G) KEGG pathway-related gene sets, showing pathways significantly correlated with C1orf198 expression in CRC.

Correlation between C1orf198 expression and immune cells in CRC

To explore the correlation between C1orf198 and immune cells in CRC, the ESTIMATE algorithm was employed (Figs. 4A–4C). Results showed positive correlations between C1orf198 expression and the ESTIMATE score (R = 0.220, P < 0.001), stromal score (R = 0.290, P < 0.001), and immune score (R = 0.108, P = 0.006), indicating its association with tumor microenvironment components. Using the ssGSEA, C1orf198 was found to significantly correlate with multiple immune cells. Notably, C1orf198 expression exhibited a correlation with macrophage infiltration (R = 0.345, P < 0.001) and M2 macrophages (R = 0.156, P < 0.001) via CIBERSORT, with similar modest effect sizes reflecting the complexity of immune cell recruitment in CRC. M2 macrophages are known to modulate multiple pro-tumor processes, including angiogenesis, extracellular matrix (ECM) remodeling, cancer cell proliferation, metastasis, immunosuppressive signaling, chemotherapeutic resistance, and reduced responsiveness to immune checkpoint blockade therapy. These associations suggest that C1orf198 may interact with M2 macrophage-driven pathways to influence CRC progression and tumor-immune crosstalk. Further analysis (Fig. 4F) demonstrated a positive correlation between C1orf198 and macrophages (R = 0.345, P < 0.001), while Fig. 4G showed its correlation with M2 macrophages (R = 0.156, P < 0.001). Associations between C1orf198 expression and macrophage markers CD163 (R = 0.360, P < 0.001) and MRC1 (R = 0.323, P < 0.001) were also observed (Figs. 4H–4I). Additionally, C1orf198 expression correlated with other immune checkpoints CD274 (R = 0.220, P < 0.001) and PDCD1 (R = 0.208, P < 0.001) (Figs. 4J, 4K), further underscoring its role in immune-cell-related processes. Collectively, these results suggest that C1orf198 may influence immune-cell-related processes in CRC, potentially impacting tumor-immune interactions.

Figure 4 The correlation between C1orf198 and immune cells in CRC.

(A–C) The ESTIMATE algorithm was used to analyze the association of C1orf198 expression with the ESTIMATE score, stromal score, and immune score. (D, E) The relationship between C1orf198 gene expression and immune cell infiltration in CRC was explored via the ssGSEA and CIBERSORT tool. (F) Correlation analysis between C1orf198 and macrophages. (G) Correlation analysis between C1orf198 and M2 macrophages. (H, I) Associations between C1orf198 expression levels and macrophage markers (CD163, MRC1). (J, K) Associations between C1orf198 expression levels and immune checkpoints (CD274, PDCD1) (ns represents no significance, * p < 0.05,** p < 0.01, *** p < 0.001).

Correlation analysis of C1orf198, cytokines and immune-related factors in CRC

Given the potential of cancer cells to modulate immune cell polarization via chemokines and their receptors, this study investigated the relationship between C1orf198 expression and chemokine/receptor profiles sourced from the TISIDB database (Figs. 5A–5C). In general, the highest correlation was observed for CXCL12 (R = 0.390) among cytokines and for CXCR4 (R = 0.367) among cytokine receptors. We thus further investigated the relationship between CXCL12, CXCR4, and immune cells, and the results were similar to those of the C1orf198 gene itself. CXCL12 significantly correlated with macrophages with ssGSEA(R = 0.541, P < 0.001) and macrophages M2 cells with CIBERSORT (R = 0.267, P < 0.001) (Figs. 5D, 5E). Similarly, CXCR4 expression strongly correlated with macrophages (R = 0.651, P < 0.001) and macrophages M2 cells (R = 0.255, P < 0.001) (Figs. 5F, 5G).

Figure 5 Correlation analysis of C1orf198 expression with cytokine families (CCL/CXCL/CCR/CXCR), associations with immune cell infiltration, macrophage markers (CD163, MRC1), and immune checkpoints (CD274, PDCD1) in CRC.

(A–C) Correlation analysis of C1orf198 gene expression with cytokines (CCL, CXCL, CCR, CXCR families) in CRC. (D, E) Correlation analysis between CXCL12 expression and immune cell infiltration in CRC. (F, G) Correlation analysis between CXCR4 expression and immune cell infiltration in CRC. (H, I) Associations between the expression levels of cytokines (CXCL12, CXCR4) and macrophage markers (CD163, MRC1). (J, K) Associations between the expression levels of cytokines (CXCL12, CXCR4) and immune checkpoints (CD274, PDCD1).

Associations between CXCL12, CXCR4 and macrophage markers (CD163, MRC1) were observed (Figs. 5H–5I). CXCL12 positively correlated with CD163 (R = 0.598) and MRC1 (R = 0.573); CXCR4 also showed significant positive correlations with CD163 (R = 0.701) and MRC1 (R = 0.626). Finally, correlations between CXCL12, CXCR4 and immune checkpoints (CD274, PDCD1) were found (Figs. 5J–5K). CXCL12 correlated with CD274 (R = 0.369) and PDCD1 (R = 0.288); CXCR4 correlated with CD274 (R = 0.579) and PDCD1 (R = 0.520), suggesting their roles in regulating immune checkpoints and tumor immune escape.

Discussion

CRC represents a biologically diverse malignancy marked by perturbed cytokine signaling networks, which disrupt multiple cellular pathways and drive tumor initiation, progression, and the acquisition of aggressive phenotypic traits (Ahmad et al., 2021). Deciphering the complex molecular mechanisms underpinning CRC pathogenesis is pivotal for advancing early detection methods, optimizing treatment regimens, and enhancing the capacity to modulate disease progression (Abedizadeh et al., 2024). A critical next step involves identifying novel biological indicators associated with immune cell infiltration landscapes and unraveling the underlying molecular pathways that dictate responses to immunotherapeutic interventions. This approach aims to deepen our understanding of CRC heterogeneity and inform the development of personalized strategies to improve clinical outcomes (Underwood, Ruff & Pawlik, 2024; Zhou et al., 2024). Notably, within the landscape of immune-based personalized therapies, immune cell PD-L1 expression is significantly higher in mismatch repair (MMR)-deficient (MSI-H) CRC compared to MMR-proficient (MSI-L) tumors, with no differences observed among distinct MSI-H molecular subtypes. Recommended screening for defective DNA MMR includes IHC and/or microsatellite instability (MSI) testing; however, challenges persist in distilling biological and technical heterogeneity into actionable data. For instance, IHC testing of the MMR machinery may yield variable results for a given germline mutation, potentially due to somatic mutations, highlighting the need for robust biomarker validation in clinical practice (Adeleke et al., 2022). Currently, researchers have successfully detected and functionally annotated nearly 90% of predicted human proteins. However, approximately 10% remain uncharacterized or poorly annotated, particularly regarding their structural features, biological functions, and roles in disease contexts. Among this understudied group are chromosome-specific open-reading frame proteins, commonly referred to as CxORFx or ORF proteins. These proteins are encoded by ORF genes, whose naming convention integrates a chromosome number (designated as “Cx”) with an open-reading frame identifier (“ORFx”), directly reflecting their genomic location (Ershov et al., 2023). Presently, increasing evidence suggests that uncharacterized ORF genes may be involved in the occurrence and development of cancer (Gao et al., 2020; Ma et al., 2019; Tang et al., 2022). As mentioned above, previous studies have shown that C1orf198 plays a role in the development and progression of gastric and breast cancers, but its relationship with CRC remains unclear.

Our study demonstrates that C1orf198 is significantly upregulated in CRC at both mRNA and protein levels, consistent across TCGA, GEO datasets, and clinical IHC samples. High C1orf198 expression correlated with aggressive clinicopathologic features and poor survival outcomes, establishing it as a potential prognostic indicator. These findings align with prior studies linking ECM and adhesion-related genes to CRC metastasis, suggesting C1orf198 may promote tumor progression by enhancing cell motility and invasive capacity. Functional enrichment analyses revealed C1orf198-associated genes were enriched in biological processes such as ECM organization, cell adhesion molecule activity, and PI3K-AKT signaling—pathways critical for tumor cell proliferation, migration, and resistance to apoptosis (Yu, Wei & Liu, 2022). The involvement of focal adhesion and ECM-receptor interaction pathways highlights C1orf198’s role in mediating cell–matrix interactions, which are pivotal for metastatic dissemination. GSEA further implicated angiogenesis and EMT, key processes in tumor invasion and TME remodeling. TME is a complex ecosystem of immune cells, stromal cells, and signaling molecules that profoundly influences cancer progression and therapeutic response. Our study revealed a significant association between C1orf198 expression and TME composition, particularly immune cell infiltration and stromal remodeling. Using the ESTIMATE algorithm, we observed positive correlations between C1orf198 levels and both stromal scores (R = 0.290, P < 0.001) and immune scores (R = 0.108, P = 0.006), suggesting a potential role in shaping the structural and immune landscapes of the TME, albeit with modest effect sizes consistent with the multifactorial nature of TME regulation. A notable finding was a correlation between C1orf198 expression and macrophage infiltration, particularly M2-type macrophages—critical mediators of tumor-promoting inflammation—though the magnitude of this association (M2 macrophages: R = 0.156, P < 0.001) indicates C1orf198 likely acts in concert with other factors to modulate immune infiltration. Analysis via the ssGSEA and CIBERSORT tool demonstrated a robust association with macrophage abundance (R = 0.345, P < 0.001) especially M2-Macropahge (R = 0.156, P < 0.001), further validated by direct correlation with macrophage markers CD163 (R = 0.360, P < 0.001) and MRC1 (R = 0.323, P < 0.001). M2 macrophages are known to secrete growth factors (e.g., VEGF) and matrix-degrading enzymes, fostering angiogenesis, ECM remodeling, and cancer cell invasion. Additionally, they suppress anti-tumor immune responses by producing interleukin-10 (IL-10) and transforming growth factor-β (TGF-β), inhibiting T-cell activation and promoting regulatory T-cell (Treg) recruitment (Li et al., 2023). The enrichment of C1orf198 in M2-related pathways suggests it may drive macrophage polarization toward an immunosuppressive phenotype, creating a niche conducive to tumor progression and resistance to immunotherapy. C1orf198 expression also correlated with immune checkpoint molecules PD-L1 (CD274, R = 0.220, P < 0.001) and PD-1 (PDCD1, R = 0.208, P < 0.001), key regulators of T-cell exhaustion. This association implies a potential mechanism by which C1orf198 facilitates tumor immune escape: upregulated PD-L1/PD-1 signaling dampens cytotoxic T-cell responses, allowing tumors to evade immune surveillance (Jiang et al., 2019). Clinically, high C1orf198 expression may predict poor response to immune checkpoint inhibitors, as observed in patients with immunosuppressive TMEs. The correlation analysis extended to cytokine-receptor networks revealed that C1orf198 was strongly linked to the CXCL12/CXCR5 axis, which is central to immune cell trafficking and TME organization (Hussain et al., 2019). CXCL12, a chemokine highly correlated with C1orf198 (R = 0.390), promotes macrophage recruitment and M2 polarization (Babazadeh et al., 2021), while its receptor CXCR5 (R = 0.367) modulates lymphocyte homing to lymphoid tissues (Elzein et al., 2021). Functional follow-up showed that CXCL12 and CXCR5 mirrored C1orf198’s associations with M2 macrophages and immune checkpoints, suggesting a coordinated role in orchestrating TME crosstalk. These interactions likely drive a feedforward loop where C1orf198 upregulates chemokine signaling, attracting immunosuppressive cells while repelling cytotoxic T cells, thereby fostering a pro-tumor microenvironment. In summary, C1orf198 emerges as a pivotal node in TME biology, integrating ECM remodeling, immune cell recruitment, and checkpoint signaling to promote a pro-tumor microenvironment. These findings deepen our understanding of CRC-immune interactions and highlight C1orf198 as a candidate for stratified therapy based on TME characteristics.

While our study provides robust bioinformatics and clinical validation, this study has several limitations that should be acknowledged. First, the sample size for clinical validation was relatively small, which may restrict the generalizability of our findings. Future studies with larger, multi-center cohorts are warranted to validate the observed associations. Second, the correlation strengths between C1orf198 and immune/stromal scores and immune cell infiltration were modest, indicating that C1orf198 is one of multiple factors influencing the CRC tumor microenvironment rather than a sole determinant. These findings highlight the need to evaluate C1orf198 in conjunction with other biomarkers to capture the full complexity of tumor-immune interactions. Finally, functional validation (e.g., in vitro/in vivo assays) is required to confirm the direct role of C1orf198 in regulating immune infiltration and oncogenic pathways.

Conclusions

In conclusion, this study identifies C1orf198 as a potential prognostic biomarker in CRC, associated with tumor progression, ECM remodeling, and immunosuppressive TME formation. The statistically significant but modest correlations observed suggest C1orf198 may contribute to CRC pathogenesis alongside other regulatory factors. Our findings highlight its potential as a candidate for further investigation as a therapeutic target, though its clinical utility should be validated in larger cohorts and in combination with other biomarkers to enhance predictive accuracy. Further functional studies are essential to unravel the precise mechanisms by which C1orf198 influences CRC pathogenesis and immune crosstalk.

Supplemental Information

Supplemental Information 1 Organize chip array design and production registration form

Supplemental Information 2 Detailed pathological data

Abbreviations The following abbreviations are used in this manuscript:

CRC Colorectal cancer

TCGA The Cancer Genome Atlas

GEO Gene Expression Omnibus

IHC Immunohistochemistry

OS Overall survival

DSS Disease-specific survival

PFI Progress free interval

TME tumor microenvironment

ECM Extracellular matrix

DFS Disease-free survival

DEGs Differentially expressed genes

GO Gene Ontology

KEGG Kyoto Encyclopedia of Genes and Genomes

GSEA Gene set enrichment analysis

MF Molecular functions

CC Cellular components

BP Biological processes

NES Normalized enrichment scores

ROC Receiver operating characteristic

AUC Area under the curve

Additional Information and Declarations

Competing Interests

Author Contributions

Human Ethics

Data Availability

The authors declare there are no competing interests.

Changjiang Yang conceived and designed the experiments, performed the experiments, analyzed the data, prepared figures and/or tables, authored or reviewed drafts of the article, and approved the final draft.

Xuhua Geng conceived and designed the experiments, performed the experiments, analyzed the data, authored or reviewed drafts of the article, and approved the final draft.

Zihan Zhao conceived and designed the experiments, analyzed the data, authored or reviewed drafts of the article, and approved the final draft.

The following information was supplied relating to ethical approvals (i.e., approving body and any reference numbers):

Supplied by Shanghai Outdo Biotech Company (Shanghai, China), the colorectal cancer (CRC) tissue microarray (HColA160CS01) comprised 80 paired tumor and ad-jacent normal tissue samples. Ethical approval for the study protocols was granted by the company’s Ethics Committee under the approval ID: YB M-05-02.

The following information was supplied regarding data availability:

The data utilized and/or examined in this research is available at Gene Expression Omnibus (GSE113513, GSE89076, GSE22598, and GSE110224) and The Cancer Genome Atlas (TCGA) network (TCGA-COAD and TCGA-READ).

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
