# Peer review of "Integrative bioinformatics analysis identifies C1orf198 as a novel prognostic biomarker in colorectal cancer"

_PeerJ, doi:10.7717/peerj.20227_

## Round 0.1 · original submission · Major Revisions

Reviewer 1 ·

Basic reporting

no comment

Experimental design

no comment

Validity of the findings

no comment

Additional comments

In this article, the authors investigated C1orf198 expression patterns, clinical correlations, functional pathways, and immune infiltration associations in CRC. The manuscript is straightforward, well written, and concise and has clear results. Definitely deserves to be published and is a valuable contribution to the “PeerJ” journal.
However, the following comments need to be addressed, as recommended.

[1] “1. Introduction”, Lines 28-29:
“Colorectal cancer (CRC), ranked among the most common malignant tumors globally, poses a significant threat to human health1.”.
From an epidemiological perspective, the authors should note that recent evidence suggests the emergence of new favorable subsets of cancers of unknown primary (CUP), including CRC CUP. This distinct clinical entity is managed as CRC and contributes to the observed increase in the current incidence of CRC.
Recommended reference: Rassy E, et al. New rising entities in cancer of unknown primary: Is there a real therapeutic benefit? Crit Rev Oncol Hematol. 2020;147:102882.

[2] “1. Introduction”, Lines 29-31:
“Although considerable advancements have been achieved in diagnostic and therapeutic approaches, treatment outcomes for a substantial portion of CRC patients still remain suboptimal2.”.
The authors should make a comment about the population of elderly patients. Please, report that even though older patients with pT4 disease are more prone to severe postoperative complications, there is no consensus that age affects survival outcomes. The prognosis of older patients may be confounded by differences in stage at presentation, tumor site, preexisting comorbidities, and type of treatment received.
Recommended reference: Osseis M, et al. Surgery for T4 Colorectal Cancer in Older Patients: Determinants of Outcomes. J Pers Med. 2022;12(9):1534.

[3] “1. Introduction”, Lines 44-46:
“Dysregulation of cell adhesion molecules, ECM-receptor interactions, and immune microenvironment has been linked to metastasis and therapeutic resistance in CRC.”.
At that point the authors should note that FOLFOX-resistance in advanced CRC is significantly associated with upregulation and downregulation of several serum miRNAs, such as miR-19a. In terms of treatment response to anti-VEGF or anti-EGFR inhibitors in metastatic CRC, upregulation of miR-126 was correlated with bevacizumab resistance, whereas overexpression of miR-31, miR-100, miR-125b, and downregulation of miR-7, with resistance to cetuximab, respectively.
Recommended reference: Boussios S, et al. The Developing Story of Predictive Biomarkers in Colorectal Cancer. J Pers Med. 2019;9(1):12.

[4] “4. Discussion”, Lines 232-234:
“This approach aims to deepen our understanding of CRC heterogeneity and inform the development of personalized strategies to improve clinical outcomes11,12.”.
At that point, the authors should mention that immune cell PD-L1 expression is significantly higher in mismatch repair (MMR)-deficient (MSI-H) CRC as compared to MMR-proficient (MSI-L) tumors, with no differences among the different MSI-H molecular subtypes. The recommended screening for defective, DNA MMR includes immunohistochemistry and/or MSI test. However, there are challenges in distilling the biological and technical heterogeneity of MSI testing down to usable data. It has been reported in the literature that immunohistochemistry testing of the MMR machinery may give different results for a given germline mutation and has been suggested that this may be due to somatic mutations.
Recommended reference: Adeleke S, et al. Microsatellite instability testing in colorectal patients with Lynch syndrome: lessons learned from a case report and how to avoid such pitfalls. Per Med. 2022;19(4):277-286.

·

Basic reporting

The article presented a topic of medical and scientific significance. The experiments were properly designed and the sources of data were clearly identified. Further, procedures for data access, handling and processing followed ethical principles and were approved. The analytical techniques were also rigorously defined.
I agree that:
1. Clear and unambiguous, professional English was used throughout
2. Literature references, sufficient field background/context were provided
3. Professional article structure was used, figures and tables were properly presented and raw data was shared
4. Article was self-contained with relevant results to hypotheses

Experimental design

One major concern is the limited sample size (80 paired tumor and adjacent normal tissue
samples) and the generalizability of the finding. Using much larger samples could support reliability and generalizability of the findings. The "R" scores were generally low making the correlations between C1orf198 expression and the ESTIMATE score (R=0.220, P<0.001), stromal score (R=0.290, P<0.001), and immune score (R=0.198, P=0.006), and the presented association with tumor microenvironment components weak and less reliable. Perhaps inputing more biopsy samples into the algorithm could improve the predictive capability of the analysis. Obviously, the only reliable and predictive correlation was between CXCR4 and CD163 with value of R > 0.70.

I agree to the following metrics:
1. Original primary research within Aims and Scope of the journal.
2. Research question well defined, relevant & meaningful. It is stated how research fills an identified knowledge gap.
4. Rigorous investigation performed to a high technical & ethical standard.
5. Methods described with sufficient detail & information to replicate.

Validity of the findings

n the same vein, it appears difficult to accept the conclusion that "Analysis via the ssGSEA and CIBERSORT tool demonstrated a robust association with macrophage abundance (R=0.268, P<0.001) especially M2-Macropahge (R=0.156, P<0.001), further validated 266, by direct correlation with macrophage markers CD163 (R=0.360, P<0.001) and MRC1 (R=0.323, 267, P<0.001)"

While the P-values are generally >0.05 and therefore suggest statistically significant difference between correlates, the low values of "R" makes the reliability of association weak. Although such low "R" values are not uncommon in biological samples' data analysis, the discussion and conclusion must admit the limited reliability associated with the outcome of the results and caution informed application to clinical decisions making, the aultimate goal of this kind of results.

Conclusions are well stated, linked to original research question but NOT limited to supporting results. Colcusions seems to overgeneralized the utility of the results.
All underlying data were provided. They are robust, statistically sound, & controlled. Discussions and conclusions did not acknowledge the liimitations of the findings.

Additional comments

In view of the above, I advise that the results, discussions and conclusions reflect the observed results with regards to the comments made in preceeding sections of this review report.

---

## Round 0.2 · accepted · Accept

Since all reviewer concerns have now been addressed, the manuscript is ready to be accepted for publication.

Reviewer 1 ·

Basic reporting

The authors have successfully addressed my comments.
The revised manuscript should be accepted for publication.

Experimental design

-

Validity of the findings

-